# siRNA Conjugated Nanoparticles—A Next Generation Strategy to Treat Lung Cancer

**DOI:** 10.3390/ijms20236088

**Published:** 2019-12-03

**Authors:** Rasha Itani, Achraf Al Faraj

**Affiliations:** Department of Radiologic Sciences, Faculty of Health Sciences, American University of Science and Technology (AUST), Beirut, Lebanon; Rasha_Itani@outlook.com

**Keywords:** lung cancer, siRNA, nanoparticles, drug delivery systems, diagnosis and therapy

## Abstract

Despite major progress in both therapeutic and diagnostic techniques, lung cancer is still considered the leading cause of cancer mortality in the world due to the ineffectiveness of the classical treatments used nowadays. Luckily, the discovery of small interfering RNA (siRNA) planted hope in the hearts of scientists and patients worldwide as a new breakthrough in the world of oncology and a robust tool for finally curing cancer. However, the valuable siRNA must be protected and preserved to ensure the effectiveness of this gene therapy, thus nanoparticles are gaining more attention than previous years as the optimal carriers for this fragile molecule. siRNA-loaded nanoparticles are being extensively investigated to find the appropriate formulation, combination, and delivery route with one objective in mind—successfully overcoming all possible limitations shown in clinical studies and making full use of this novel technique to become the next generation treatment to wipe out many chronic diseases, including cancer. In this review, the benefits of using siRNA and nanoparticles in lung cancer treatment will be globally reviewed before discussing why and how nanoparticles and siRNA can be combined to achieve an efficient treatment of lung cancer for prospective clinical applications.

## 1. Introduction

According to the World Health Organization (WHO), lung cancer is still considered the leading cause of death in both men and women worldwide with approximately 2.09 million reported cases and 1.76 million deaths in 2018 [1]. Multiple factors, both environmental and man-made, lead lung cancer to be the primary cause of death. These factors include smoking and/or second hand smoke of cigarettes and hookah (i.e., their smoke contains over 60 carcinogens such as radioisotopes from Radon decay sequence, Nitrosamine, Benzopyrene, etc. [2,3]), air pollution (i.e., fine particulates and aerosols released from car exhausts and factories), exposure to toxins such as Asbestos (i.e., can also cause lung pleura cancer—mesothelioma), Arsenic, and Radon gas, etc. [4]. Figure 1 summarizes worldwide lung cancer statistics and some of the various risk factors that mainly contribute to this health-threatening disease.

However, what lead to the high death toll for lung cancer is not only the disease itself but also its late diagnosis where lung cancer symptoms (i.e., cough, hemoptysis, dyspnea, pneumonia) appearing in late stages limit the treatment options. In addition, most lung cancer cases are detected coincidentally in routine chest X-ray or Computed Tomography (CT) examinations due to the absence of regular or annual screening for lung cancer in patients above 50 years with a 30-year smoking experience [6].

Lung cancer can be classified into as either Small Cell Lung Cancer (SCLC) or Non-Small Cell Lung Cancer (NSCLC), with NSCLC being the most common form (85% of reported cases) [7]. Both can be screened using a variety of imaging such as conventional X-ray (chest radiography), Computed Tomography (CT), Magnetic Resonance Imaging (MRI), and Positron Emission Tomography—CT hybrid (PET/CT). Other screening tools involve bronchoscopy where a physician inserts a specially designed endoscope through the nose or mouth to the lungs with the patient under sedation, the images are observed in real-time on a screen. This is a diagnostic and therapeutic tool since it can allow the confirmation of the presence or absence of a tumor, sampling, and/or resection of the tumor [8].

Classical treatments of lung cancer include surgery, chemotherapy, and radiation therapy, each adapted to suit the patient’s needs and condition. However, at advanced stages, and especially metastatic stages (i.e., stage IV for NSCLC and extensive stage for SCLC), treatment options may be limited to chemotherapy, which has numerous pitfalls. The problem with most chemotherapy treatments is the fact that most are platinum-based and use medications such as Cisplatin (Platinol) and Carboplatin (Paraplatin) which can be very harmful to the body causing many side effects (including pain, blood clots, trouble breathing, bone and dental issues, anemia, cardio- and nephro- toxicity, neuropathy, hair loss, fatigue, weakness, etc.) that can degrade the quality of life of cancer patients. Moreover, chemotherapy is not cell-targeted (specific). As a consequence, the use of a high therapeutic dose of anti-cancer drugs will not be effective thus limiting the treatment [9].

Fortunately, new and innovative treatments for lung cancer are emerging with the aim of improving the quality of life for cancer patients and providing alternatives to classical treatments with many side effects like chemotherapy. One new strategy involves exploiting the unique characteristics of nanoparticles not only to accurately deliver the anti-cancer medications to the tumor site, but also to target small interfering RNA (siRNA) to be used as a gene silencing tool to suppress the expression of various genes including those that encode for lung cancer. Each of these new methods has its own advantages and complications that stand in the way of its clinical use.

This review article will discuss, in depth, three major topics: (1) how siRNA can be used for treating lung cancer, its mechanism, and challenges, (2) the different types of nanoparticles used in the delivery of therapeutic drugs, and finally (3) siRNA conjugated nanoparticles types, effects of changing the nanoparticles’ structure on siRNA delivery, the three main delivery routes, and the many barriers of siRNA delivery to the lungs.

## 2. Small Interfering RNA (siRNA) for Lung Cancer Therapy

### 2.1. Background

RNA interference (RNAi) is a new and powerful gene silencing tool that could be the next generation solution for treating all lung cancer types and stages [10]. The initiation of RNAi depends on many types of small RNAs which play an important role in degrading messenger RNA (mRNA). The clinical applications and success of RNAi-based therapies heavily depend on their delivery mechanism: finding an appropriate delivery route, delivery carriers, and their unlimited modifications and conjugations for precise and safe delivery.

### 2.2. Mechanism of Action

RNAi is a gene silencing tool used to suppress the expression of many genes by using small interfering RNA (siRNA), short hairpin RNA (shRNA), and micro RNA (miRNA). All these RNAs are cleaved by an enzyme called Dicer that allows the modification of genes by providing transcription factors. The cells are exposed to long double-stranded RNAs (dsRNAs) that will be processed to smaller fractions forming complexes with RNA-induced silencing complexes (RISCs). The sense strand will, later on, be cleaved by Argonaute 2 (AGO2) and the anti-sense strand will guide the RISCs towards the complementary mRNA divided by AGO2 into two mRNA fragments. The application of siRNA at this level will help downregulate the target genes without starting any responses [11].

### 2.3. Challenges

Despite its great and promising potential for treating severe diseases including cancer, naked siRNA administration still faces many barriers that stand in the way of its clinical use. Factors such as rapid enzymatic degradation, short half-life, weak permeability to cell membranes due to their negative charge, instability, removal by glomerular filtration, etc., all limit the use of locally administered siRNA agents [12]. Therefore, the conjugation of siRNA to specially designed, biocompatible carriers and their chemical modification to overcome all the physical and inherent barriers are essential for the success of the treatment.

Out of the various candidates used for siRNA delivery were viral vectors. Their effective and accurate delivery of nucleic acid molecules and their ability to protect these acids from degradation and elimination raised hopes for improving local and systemic RNAi-based therapies. However, clinical trials uncovered the multiple drawbacks which limited their application—these viral carriers are unable to deliver sufficient amounts of RNAi and can easily provoke immune responses. As a consequence, intensive research is being carried out to find stable, biodegradable, and biocompatible vectors that can safely carry these nucleic acids to their target sites without initiating an immune response.

Nanoparticles are now emerging as effective carriers due to their low toxicity, size, charge, and chemical modification capabilities that allow them to overcome the multiple barriers that stood in the way of their previous counterparts. They are a promising tool for the next generation treatment of various diseases including lung cancer.

## 3. Nanoparticles for the Delivery of Therapeutic Drugs and Compounds

### 3.1. Background

Nanotechnology unshed a new age in lung cancer therapy due to their ability to target and accurately deliver anti-cancer drugs and agents to the tumor site. The basis of this therapy involves the encapsulation of these drugs to specific nanoparticles that serve as nanocarriers to precisely deliver these drugs and reduce the unwanted side effects by altering their biodistribution and pharmacokinetic properties [13].

Compared to traditional chemotherapy treatments that utilize free drugs, nanoparticle-mediated therapies allow the delivery of hydrophobic/hydrophilic drug molecules, peptides, radionuclides, antibodies, etc. to the tumor site safely and accurately. Its advantages include lowering the dose of the therapeutic agents, decreasing unwanted side effects and drug-related toxicity, improving the plasma half-life of the drugs, enhanced permeability and retention (EPR) effect, and controlled drug release. Moreover, the modification of the physical and chemical properties of the encapsulated drugs leads to increased tumor localization and improved tumor response to the drugs [14].

Recently, various nanoparticles have been specifically tailored and designed to deliver anti-cancer drugs and nucleic acids like DNA and RNA to lung cells thus opening new doors in lung cancer therapy strategies [15]. These nanoparticles that can be classified into either organic or inorganic (Figure 2) will be briefly described in the following sections.

### 3.2. Lipid-Based Nanoparticles

The two most important classes of lipid-based nanoparticles for drugs delivery include liposomes and micelles. Liposomes are bi-layered vesicles or sacs used to carry and deliver anti-cancer agents where the hydrophilic agents are encapsulated in the inner aqueous core and the hydrophobic agents are integrated into the phospholipid bilayer. The structure of these lipid-based nanoparticles and their composition can be adjusted to change the physical and chemical properties such as size, shape, and charge to enhance their effects, tumor localization, and prolong the blood circulation time [16].

Lipid-based nanoparticles are non-toxic, biocompatible, non-immunogenic, and soluble agents that are becoming a favorable platform for delivering therapeutic agents. Polyethylene glycol (PEG)-coated liposomes, also called “Stealth” liposomes, are stabilized and have increased the in vivo circulation half-life from few hours up to 45 h approximately due to reduced uptake by the reticuloendothelial system (RES) and phagocytosis which can decrease the amount of anti-cancer drugs delivered to the target site, thus limiting the treatment [17]. Several PEGylated liposomes are being recently tested, such as Doxil^®^, DaunoXome^®^, DepoCyt^®^, and ONCO-TCS^®^. They involve liposomes conjugated to different chemotherapeutic drugs and managed to reach phase II trials [18].

Focusing on NSCLC, lipid-based nanoparticles play an important role in reducing the side effects of Cisplatin and improve the efficiency of the therapy. A randomized Phase III study on Lipoplatin in the treatment of NSCLC compared the results, responses, and toxicities of Lipoplatin + Paclitaxel versus Cisplatin + Paclitaxel used as a first-line treatment. Results of that study showed that there is an increase in tumor response rate in the Lipoplatin group (59.22%) compared to the Cisplatin group (42.42%) in addition to reducing the side effects of Cisplatin [19].

Another class of lipid-based nanoparticles are micelles that have a hydrophobic core filled with therapeutic agents and a PEG hydrophilic shell. Their distinguishing properties involve long bloodstream circulation, high binding specificity to target cells, and reduced side effects [20]. Micelle formulations containing Doxorubicin, Paclitaxel, SN-38, Cisplatin, and Platinum II are undergoing clinical trials with some advancing to Phase II studies. They have proved their effectiveness against various tumors and reduced side effects making them promising for clinical use [21].

### 3.3. Polymer Nanoparticles

Polymer-based nanoparticles occupy a huge part in nanomedicine and involve many types such as polymer micelles, dendrimers, polymersomes, polymer-lipid hybrids, etc., all used to improve the efficiency of cancer therapies. FDA-approved albumin-based nanoparticle carrying Paclitaxel used as a first-line treatment of advanced NSCLC in combination with Carboplatin in inoperable patients was reported to improve the efficiency of chemotherapy in both in vitro and A549 xenograft model of lung tumor [22].

Moreover, PEG-poly glutamic acid block co-polymer micelles with encapsulated Cisplatin showed prolonged blood circulation time and accumulation in solid tumors 20 times higher than Cisplatin in its free state due to the ability to release its cargo after accumulation at the delivery site. Treatments with polymer micelles have shown complete tumor shrinkage and disappearance with no reported weight loss compared to 20% weight loss for free-roaming Cisplatin [18,23,24].

Gelatin nanoparticles containing biotinylated epithelial growth factor (EGF) molecules for lung cancer targeting showed elevated uptake in A549 adenocarcinoma cells of the lungs on trials performed on mice thus inhibiting tumor growth [25,26].

Polymeric micelles have high thermodynamic and favorable kinetic properties that allow specific delivery of anti-cancer agents to tumor sites. Micelles containing Cisplatin and Paclitaxel can release these drugs only upon the accumulation of these nanoparticles at the tumor site thus reducing toxicity and enhancing tumor response to therapy [23,27,28].

### 3.4. Dendrimers

Dendrimers are synthetic nanoparticles having a tree-like structure due to the high number of extensions and repeated branching. They have an easily modifiable surface that allows the encapsulation of anti-cancer drugs either in the core or on their surface. Polyglycerolsuccinic acid dendrimers containing encapsulated Camptothecin studied in NSCLC mouse model showed that they were taken up by the adenocarcinoma cells effectively [29].

The main characteristics of dendrimers include extremely high stability, water solubility, and decreased antigenicity. Therefore, they can be used for many applications including active and passive targeted drug delivery in modified chemotherapy treatments, gene delivery in newer cancer therapies that involve gene silencing techniques, and even as contrast agents for MRI [30].

Polyamidoamine dendrimers carrying Doxorubicin using pH-sensitive and -insensitive linkers and combined with multiple photochemical internalization (PCI) providers showed improved efficiency, reduced side effects of the anticancer drugs, and improved toxicity of Doxorubicin by increased accumulation [31,32,33].

### 3.5. Inorganic Metal-Based Nanoparticles

Metal-based nanoparticles such as Gold, Silver, Platinum, Iron Oxide, Quantum Dots, etc., have many promising biological and biomedical applications and are currently under intensive investigation to exploit the properties that make them good candidates for clinical applications in the future.

One of the most important metal-based nanoparticles is Gold nanoparticles due to their resonance-enhanced properties that allow the diagnosis and therapy of lung cancer and differentiating lung cancer histologies: normal and cancerous cells, NSCLC and SCLC including their sub-divisions [34]. As shown in Figure 3, Gold nanoparticles have a three-in-one action allowing multi-modality imaging, accurate and safe delivery of cargo, in addition to therapy of multiple diseases with techniques like thermal ablation. Thanks to the advancements in nanomedicine, these metal-based nanoparticles can be modified and conjugated to functional and chemical groups, radionuclides, and biological molecules which increase their diagnostic and therapeutic utilities. Gold nanoparticles combined with Methotrexate (MTX) proved high tumor uptake and increased therapeutic efficacy [35].

Moreover, Silver nanoparticles also play an important role in nanomedicine as they can be used as antiviral, antibacterial, antifungal, and most importantly, anti-cancerous agents. The combination of Silver nanoparticles with other therapeutic anti-cancer agents allows overcoming the tumor resistance and side effects that limited their usage. As an example, the chemotherapeutic drug 5-FU is extremely toxic and has low activity in the target cells (tumor cells). The combination of Silver nanoparticles and 5-FU showed enhanced anti-tumor response with reduced unwanted side effects as the drug was specifically delivered to the tumor site [36].

Another application of metal-based nanoparticles involves the photo-thermal killing of cancer cells by amplifying the effects of low laser radiation and exploiting the altered biomarkers of tumor cells. Silver and Gold nanoparticles can greatly amplify the thermal lethality of cancer cells by their accumulation at the tumor site and enhancement of the effects of x-rays and heat thus preventing the cells’ division by promoting apoptosis [37].

In addition to that, metal-based magnetic nanoparticles are studied and proved to be successful in both the diagnosis and treatment of various cancer types including lung cancer. Super Paramagnetic Iron Oxide (SPIO) nanoparticles were reported to generate lethal heat when exposed to alternating magnetic field gradients proved to be beneficial in the destruction of cancer cells in a mouse with NSCLC [38].

## 4. siRNA Conjugated Nanoparticles for Lung Cancer Therapy

### 4.1. Background

As discussed earlier, the direct injection of naked, unmodified siRNA was not found to be highly effective as siRNA will suffer from many challenges such as RNA degradation, very short half-life and circulation time, weak targeting and biodistribution, etc. Therefore, in order to enhance its therapeutic efficacy and make full use of its capabilities and unique characteristics, siRNAs should be encapsulated in special delivery carriers—the nanoparticles.

It is necessary to note before discussing siRNA-loaded nanoparticles the fact that many scientists have tried modifying the siRNA before injecting it in hopes of enhancing its therapeutic effect and overcoming the barriers that stop it from reaching clinical applications. Chemical modifications performed on siRNA backbone, especially at the 2′ Ribose sugar, allow activity enhancement and prolonged half-life without function alteration. Moreover, modifications done at some linkages such as phosphorothioate and boranophosphate can lead to increased efficiency of siRNA [40]. However, despite all efforts in modifying siRNA to overcome all possible barriers, it remains poor and unable to succeed in clinical trials due to the fact that excessive modification can alter the molecule’s function and biodistribution.

There are countless modifications that can be performed to the siRNA molecule, touching the backbone, sugar (ribose), phosphates, bases, nucleotides, etc. For instance, substitutions in the ribose sugar can help the siRNA molecule overcoming degradation. However, replacing about 50% of nucleotides, especially in the 2′ position, can severely interfere with the silencing process by increasing the molecule’s thermal stability. As a consequence, this can prevent the dsRNA that were broken down into simpler and smaller fractions from forming complexes with RISCs. Therefore, the mRNA will not be divided and the silencing or downregulating action of siRNA will be inhibited [41]. Nanoparticles are emerging as fascinating tools not only for bio-imaging (diagnosis) as previously considered but also for therapy by acting as specific carriers to accurately and safely deliver siRNA to the appropriate target sites. However, not all existing nanoparticles are ideal candidates for this mission as they must possess the correct formulation, length, size, shape, charge, etc., and must be successfully able to: (1) protect and fortify the cargo—siRNA, (2) have low toxicity, (3) avoid triggering an immune response, (4) accurately deliver and release the cargo at the target site, and (5) preserve the cargo’s physiology and function [42].

### 4.2. Effects of Nanoparticle Modification on Delivery of siRNA

Since nanoparticles should meet specific rules or conditions to be able to successfully deliver siRNA to the target site—which can seem impossible to satisfy—their formulation has to be modified to suit these conditions and boost the success rate of the treatment. The nanoparticles’ modifications will be briefly discussed.

Size and Shape: Size plays an important role in the nanoparticle’s pharmacokinetic behavior where medium-sized particles about 50 nm in diameter were reported to have the highest levels of cellular uptake compared to nanoparticles having a 14 or 75 nm diameter [43].

In addition, nanoparticles’ shape greatly affects the cellular internalization. In a study comparing round to rod-shaped nanoparticles, round nanoparticles were found to have five times more cellular internalization compared to their rod-shaped counterparts. This is due to the fact that the time and effort required for the cells to completely wrap up the rod-shaped nanoparticles by their membranes and absorb them was far more than the round nanoparticles [44]. Relating what has been reported to lung cancer therapy, most nanoparticles are deposited in the lungs through mechanisms like sedimentation and diffusion due to the lung’s physiology. As a consequence, nanoparticles having a larger aerodynamic diameter (D_a_) > 5 μm are deposited in the upper airways (far away from tumors in deeper segments and lobes) compared to those having a smaller diameter 1 μm < D_a_ < 4 μm that are deposited at deeper parts of the airways. However, this does not mean that the smaller the size the better since particles with a diameter <1 μm get exhaled outside the lungs [45,46,47,48].

Charge: Surface charge is another important factor that affects nanoparticle’s pharmacokinetic properties. The use of either negatively or positively charged nanoparticles is heavily case-dependent. From one side, cell membranes are usually negatively charged, thus negatively charged nanoparticles tend to repel and will be unable to move inside the cells or even cross their membranes. In this case, positively charged nanoparticles will be preferred to deliver the siRNA [49]. Studies involving Poly Lactic acid—Poly Ethylene Glycol (PLA-PEG) nanoparticles covered with stearylamine—a cationic (positively charged) lipid—showed increased internalization in HeLa cells compared to PLA-PEG nanoparticles without stearylamine coating (i.e., without a positive charge) [50]. From the other side, cationic nanoparticles might cause problems such as reducing membrane integrity, damaging mitochondria and lysosomes, etc. Therefore, anionic nanoparticles will be preferred in these conditions. Moreover, phagocytic cells prefer interacting with anionic nanoparticles more than their positively charged nanoparticles [51].

It is also crucial to note that nanoparticle size, charge, and shape are not the only conditions one should pay attention to. Experimental conditions such as the buffer used, the medium in which the experiment was conducted, the temperature, and the pH are also important factors [52]. In most cases, even a simple variable can be game-changing: Carboxy (PS-COOH) and amino-functionalized polystyrene (PS-NH_2_) having the same size (100 nm) but conducted in different experimental conditions showed different internalization methods [53]. In addition to that, the interaction between nanoparticles and cell membrane not only depends on the nanoparticle size, but also on the membrane wrapping process that starts off endocytosis. Therefore, nanoparticles having a small size and less receptor—ligand interactions need to be close enough to the cell to initiate membrane wrapping and vice versa [54].

Hydrophobicity is another important factor that should not be ignored. For some cells, hydrophobic nanoparticles can be stuck in the bi-layer, whereas semi-hydrophilic nanoparticles can be absorbed into the membrane [55]. In another study, dichain nanoparticles modified by DMAB showed greater interaction and cellular internalization than their single chain versions. This is due to the fact that having two hydrophobic chains lead to more interaction than single chains CTAB (Cetyl-Trimethyl-Ammonium Bromide) and DTAB (Dodecyl-Trimethyl-Ammonium Bromide) [56]. Therefore, the choice of hydrophobic, hydrophilic, and/or semi-hydrophobic nanoparticles heavily depends on the cell type.

### 4.3. Types of Nanoparticles Used for siRNA Delivery to the Lungs

#### 4.3.1. Organic Nanoparticles

##### Lipid-Based Nanocarriers

They include all cationic (positively charged) lipid nanoparticles, lipid-like substances, and liposomes. What makes lipid-based nanoparticles good candidates for siRNA delivery is the fact that they are effective carriers that can be easily modified and functionalized. Moreover, lipids are known for their good interaction with the negatively charged cell membranes. However, one major drawback of these nanoparticles for prospective clinical applications is their biocompatibility but not their inherent toxicity: many researched siRNA loaded lipid-based nanoparticles used for the treatment of lung cancer have not been approved and failed to be commercially available due to their toxicity caused by the indirect activation of cytokine. Two of the aforementioned compounds are Oligofectamine and Lipofectamine [57].

##### Polymer-Based Nanocarriers

Polyethyleneimine (PEI) polymers are considered the best polymer-based nanoparticles for siRNA delivery thanks to their positive charge, molecular weight, and special branching pattern. PEIs have unmatched transfection efficiency but they are also limited in use due to their cellular toxicity in many cell types. Fortunately, this toxicity can be reduced by the addition of hydrophobic, hydrophilic groups, or both. Experiments involving PEIs modified with both hydrophobic and hydrophilic groups have shown reduced cytotoxicity in mouse models using intratracheal delivery method [58].

Dendrimers are also used for siRNA delivery due to their various nanoparticles—cargo binding mechanisms which include adsorption, chemical conjugation, and encapsulation. Polyamidoamine (PAMAM) dendrimers have been studied on a large scale for many years as a potential carrier for siRNA. PAMAM advantages include the easy binding to siRNA thanks to their surface amines and also the efficient escape of the cargo at the appropriate target site(s) with the help of their internal amines. A study on modified PAMAM showed less densely packed dendrimers when PAMAM had a triethanolamine core leading to increased internalization of siRNA [59].

#### 4.3.2. Inorganic Nanocarriers

Inorganic nanocarriers have fascinated scientists due to their wide potentials. They are not only limited to metallic nanoparticles but also involve many sub-types such as semiconductor nanoparticles, Carbon-based nanoparticles, silica nanoparticles, quantum dots, fullerenes, etc. These nanoparticles have long astonished researchers and scientists due to their two-in-one role as both carriers for siRNA for lung cancer therapy and bio-imaging tools used in diagnosis and accurately tracking the siRNA trajectory upon delivery and its activation at target sites. Despite the debate regarding the toxicity of metal-based nanoparticles, new studies have shown that they can be modified like other nanoparticles easily to reduce their toxicity levels. Moreover, their benefits that include stability, noninvasive fluorescent nature, and controllability might outweigh their disadvantages and drawbacks [60].

##### Gold Nanocarriers

Gold nanoparticles have been studied extensively throughout the years as the ultimate candidates for siRNA delivery. Their special surface plasmon resonance (SPR) characteristic makes them beneficial for bio-imaging, and in addition to their stability and efficient delivery of siRNA, a lot of effort is being put to exploit the true potential of these particles and further improve their biocompatibility for prospective clinical applications [61].

##### Iron Oxide Nanocarriers

Many metal oxide particles, especially iron oxide particles, have taken the spotlight due to their unique characteristics. In addition to the fact that these nanoparticles can be used for bio-imaging, have excellent cellular absorption, are stable and modifiable like other metal-based nanoparticles, they have the distinguishing feature of thermal activation that is also shared with gold nanoparticles. Iron oxide particles can be used to heat the tumors to lethal temperatures causing the coagulative necrosis of tumor cells upon the application of alternating magnetic fields [62]. Furthermore, SPIO nanoparticles were reported to be perfect candidates for future siRNA therapies for many diseases including lung cancer given their strong contrast (i.e., MRI signal) and their unique magnetic properties which allow them to be guided using an external magnetic field to accumulate in the tumor sites [63]. Magnetic targeting can improve both the delivery of siRNA and/or therapeutic compounds (i.e., chemotherapeutic drugs) to improve cancer treatment [64]. We have previously reported that targeting of intravenously injected SPIO nanoparticles to the lung was proved to be enhanced when using external high-energy magnets positioned over a specific region of the lung [65]. This approach was further elaborated in another study in which the use of high-energy magnets offered improved theranostic effect of Doxorubicin-loaded iron-tagged nanocarriers, by magnetically targeting them towards metastatic tumor sites in the lungs [66].

### 4.4. Delivery Mechanisms of siRNA Loaded Nanoparticles for Lung Cancer Treatment

#### 4.4.1. Intratracheal Delivery

It is one of the most widely used methods for drug delivery to the lungs. In general, most studies on siRNA-loaded nanoparticles carried on animal models use intratracheal siRNA delivery techniques, however, it is not used in humans. Although this technique provides negligible loss of therapeutic material, high efficiency, promptness, and low cost, but its major drawback is the surgical procedure that has considerable risk and is uncomfortable for the patient [67]. As an example, PEG-coated nanoparticles modified by Arginine-Glycine-Aspartic acid (RGD) peptide loaded with mouse c-myc siRNA administered through intratracheal route showed successful downregulation of c-myc gene expression and stopped tumor proliferation with little loss of therapeutic material [68].

#### 4.4.2. Intranasal Delivery

What makes intranasal delivery superior to intratracheal delivery is that it can be used in humans. In fact, many commercially available drugs are being used in humans to treat diseases like asthma and respiratory infections and are available in the form of sprays and nasal droplets. Delivery of siRNA through these devices is painless, however, humans cannot be compared to the mice or rats used in pre-clinical studies. The reason behind that is the fact that animals breathe mostly through their nose compared to humans, in addition to that, the lung anatomy and pathway differ between humans and animals so the quantity of therapeutic material delivered to the target site through this method will be relatively low. An experiment conducted on normal adult volunteers showed that upon inhalation of mono-disperse particles via intranasal delivery, only 3% of the particles reached the lungs and the remaining 97% remained in the nose due to them being captured by the nasal hairs and cilia [69].

#### 4.4.3. Intravenous Delivery

Although it possesses multiple problems and challenges, intravenous siRNA-loaded nanoparticles remain the most practical and most applicable method for delivery in humans. The problem is that upon administration of this compound, it will not reach the target organ or site directly and at the same amount as expected, instead, it undergoes multiple passes and circulations in the body and distributes unevenly in multiple organs leading to little accumulation in the target organ or site. Also, during its circulation, the compound undergoes filtration and elimination by the liver and excretion by the kidneys which shorten the circulation half-life and reduce the efficiency of the treatment. Fortunately, researchers have proposed a “trick” to overcome this problem. The secret lies in the use of sticky siRNA (ssiRNA): in vivo injection of ssiRNA and PEI successfully lead to the downregulation of tumor proteins, blockage, and prevention of tumor growth [70].

The strengths and pitfalls of each delivery method previously discussed are summarized in (Figure 4).

### 4.5. Barriers of siRNA-Loaded Nanoparticles Delivery to the Lungs

Nanoparticles carrying siRNA must overcome multiple barriers both extracellular and intracellular to prove themselves worthy of becoming the next generation therapy tool for the treatment of multiple chronic and fatal diseases including lung cancer.

Extracellular barriers can be classified as biological, chemical, and physical barriers which are all related. One of the many barriers includes opsonization where foreign particles originating from outside the body called “antigens” are destroyed by the immune system by phagocytes and macrophages. Moreover, siRNA conjugated nanoparticles delivery to the lungs is more challenging compared to other organs due to the high number of immune cells (phagocytes and macrophages) that are proliferating in the lungs. It is worth noting that that the average human breathes more than 1,000,000 viruses and airborne bacteria, in addition to the toxic particles due to air pollution, therefore, the lungs are exposed to a wide variety of antigens and must be reinforced and protected more than any other organ by macrophages and phagocytes. One proposed method to overcome this issue involves either making the injected therapeutic compound somehow “invisible” to the immune cells by modifying the surface of the nanoparticles or decreasing the interaction between the nanoparticles holding the siRNA and other particles.

Intracellular barriers involve the inaccurate delivery, escape or leak, or inadequate packing of siRNA into the nanoparticles. Endocytosis poses a big problem that stands in the way of siRNA accurate delivery where the nanoparticles containing the siRNA can be phagocytosed and released at random sites different from the desired site. Moreover, the reticuloendothelial system (RES) can pose a threat to the accurate delivery of siRNA via the nanoparticles whereupon administration, these particles are directly transported to the RES organs such as spleen and liver to be later excreted by the kidneys in the body’s natural act of removing all unwanted antigens from the body that could be harmful. Therefore, the siRNA accumulates in the liver and spleen instead of the target organ and could escape the nanoparticle capsule upon phagocytosis due to inappropriate packing [71].

## 5. Conclusions and Future Perspectives

The discovery of siRNA broke all the limitations and opened new doors for the treatment of a wide variety of diseases. This innovative gene silencing tool is expected to be far more superior to any other treatment for lung cancer due to its ability to stop the cancer cells at their origin by preventing mRNA translation. However, despite its huge potential, siRNA is weak on its own and cannot overcome the multiple barriers that stand in the way of its delivery. Fortunately, scientists have proposed the use of nanoparticles as the optimal delivery system to overcome the challenges this valuable siRNA might face. For prospective safe clinical applications, it is now time to make full use of their power, and as scientists and researchers are digging deeper into the well of nanotechnology, cancer therapy can finally take a more precise route—personalized medicine. It is now possible, after many years, to do so, due to the diversity of nanoparticles, the hundreds of siRNA—nanoparticle combinations, and the various modifications that can be done to each of siRNA or nanoparticle’s structure to match the tumor’s heterogenicity and diversity. Therefore, with siRNA-loaded nanoparticles, oncology can become personalized, thus saving millions of lives of patients with late stage cancer and improving the lives of other patients undergoing traditional treatments like radiotherapy and chemotherapy. In addition to that, the robust nanoparticles can allow co-delivery of therapeutic compounds to achieve a synergetic effect. Due to the nanoparticles’ abilities to overcome the multiple barriers and drug resistance mechanisms, they can be used to deliver both siRNA to silence the genes controlling drug resistance and also chemotherapeutic compounds to completely eliminate the disease [72]. Therefore, this novel combined therapy will exploit each and every molecule’s power to fight the tumor from all sides (i.e., genetically and chemically). However, the challenge lies in finding the optimal conjugations and pass all clinical trials and get the FDA approval. Only after that, scientists and researchers can declare finding the ultimate cure for cancer: a next generation therapy tailored to meet every patient’s case and overcome all its barriers.

## Figures and Tables

**Figure 1 ijms-20-06088-f001:**
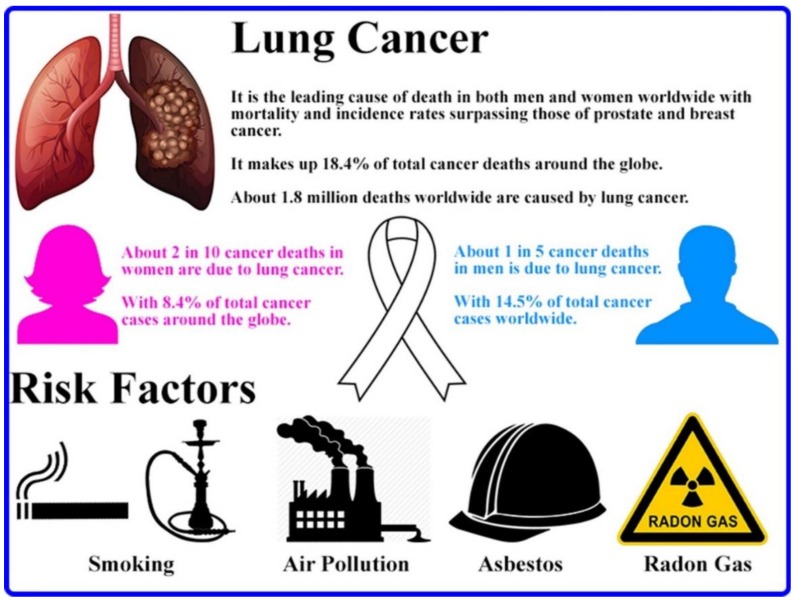
Scheme summarizing worldwide lung cancer statistics and presenting its main risk factors [5].

**Figure 2 ijms-20-06088-f002:**
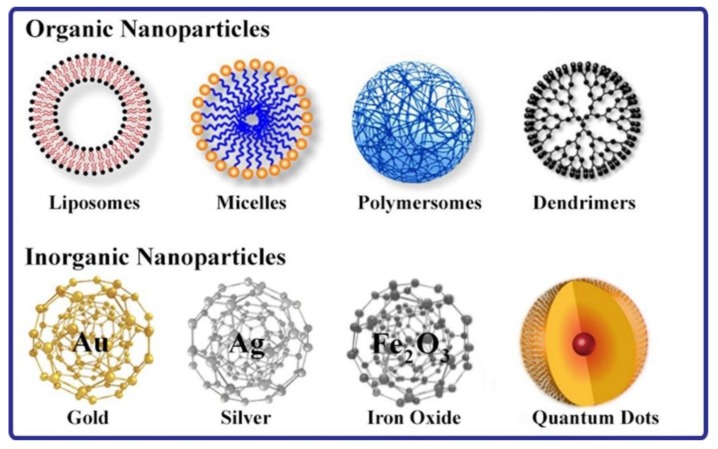
Scheme showing the different types of nanoparticles used in drug delivery for the treatment of the various types of cancers.

**Figure 3 ijms-20-06088-f003:**
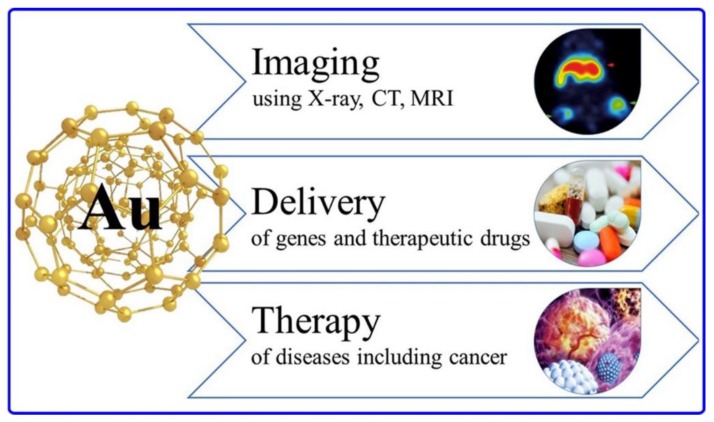
Schema showing some applications of Gold nanoparticles in diagnosis, imaging, and therapy of various cancer types. Adapted from Reference [39].

**Figure 4 ijms-20-06088-f004:**
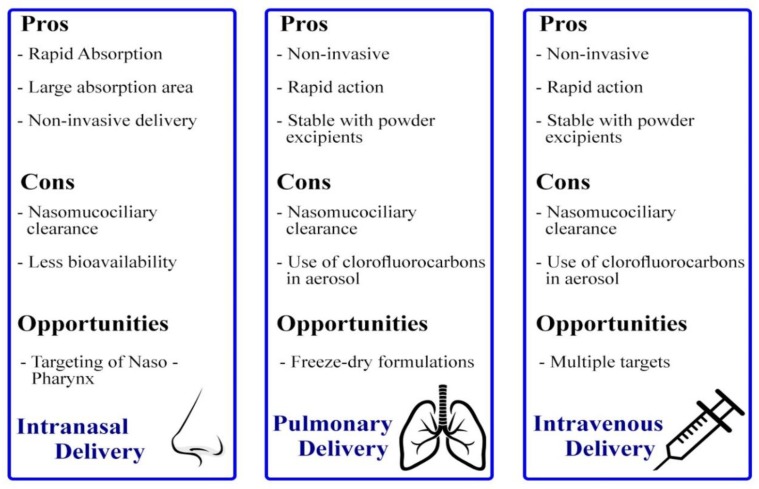
Scheme showing the advantages, disadvantages, and opportunities of some delivery methods for siRNA loaded nanoparticles to the lungs. Adapted from Reference [42].

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
