# Peer review of "siRNA Conjugated Nanoparticles—A Next Generation Strategy to Treat Lung Cancer"

_ijms, 2019, doi:10.3390/ijms20236088_

Round 1

Reviewer 1 Report

This is very interesting and well-organized review manuscript. 

Few comments/suggestion:

some figures are repeated in the file of the manuscript - please check
authors are encouraged to create the table that will summarize the application of siRNA Conjugated Nanoparticles this will be more legible for further readers.

Author Response

Response to Reviewer 1 Comments

This is very interesting and well-organized review manuscript.

The authors would like to thank the reviewer for the valuable comments. 

Few comments/suggestion:

Some figures are repeated in the file of the manuscript - please check

Response 1: Figures were directly inserted into the main text, as per the “Instructions for Authors” guidelines, and the manuscript was uploaded to the submission system. We have carefully double checked that there is no duplication in the figures.

Authors are encouraged to create the table that will summarize the application of siRNA Conjugated Nanoparticles this will be more legible for further readers.

Response 2: The authors would like to thank the reviewer for this comment but would like to kindly note that the current review article focuses only on the treatment of Lung cancer using siRNA-conjugated nanoparticles. The reviewer is invited to read the following review articles discussing several applications of siRNA-conjugated NPs: Kim YD et al., Curr Pharm Des. 2015; Young SW et al., Crit Rev Oncol Hematol. 2016; Chen X et al., Cancer Metastasis Rev. 2018.

Reviewer 2 Report

The paper reviews recently developed siRNA conjugated nanoparticles for lung cancer treatment. This is an important research field and has tremendous clinical potential. The paper is informative, well-written, although a better organization and further search on recent publications are suggested. After all, the manuscript might require a major revision before acceptation. My suggestions are listed below.

1, The authors used long paragraphs to discuss using nanoparticles for therapeutic drug delivery, I understand they just want to give a more detailed background on how important nanoparticle-based delivery systems are. However, I would suggest the authors shorten this part as much as possible so they don’t lose the real focus of the paper.

2, In section 4.1, line 260, the authors discussed that the excessive modification of siRNA could lead to the change of a molecule’s function and biodistribution. Could the authors give some examples or literature evidence here? How is the modification going to change the bio-functions of siRNA? And what are the exact consequences of doing so?

3, In section 4.2, line 274, the authors concluded that nanoparticles with average 50-nm in size have the highest cell uptake efficacy. This is case-dependent the optimum size varies significantly with the type of the particle, surface charge and properties (e.g. hydrophobic or hydrophily), and the cell type (please check J. Nanobiotechnology, 2014, 12, 5). I would suggest the authors give more detailed discussions here.

4, In section 4.2, line 289, the authors wrote that ‘negatively charged nanoparticles will be repelled and unable to move inside the cells…’. However, I believe this conclusion is not correct. There is no double that positively charged nanoparticles are more favorable for cell uptake, but numerous researches are reporting that negatively charged NPs could also penetrate cells. For example, He et al found that NPs with slight negative charges and particle size of 150 nm were tended to accumulate in tumor more efficiently (Biomaterials, 2010, 31, 3657). The authors are encouraged to check on some important references and revise this part (e.g ‎Int. J. Nanomed, 2012, 7, 5577; Biomaterials, 2003, 24, 1001)

5, One additional important advantage of using iron oxide particles for delivery is they can be delivered to targeted site via an external magnetic field, which can dramatically enhance the therapeutic efficacy. Perhaps the authors could discuss this point in the manuscript.

6, The nanoparticles based co-delivery of anti-cancer drug and siRNA is an important route to enhance the therapeutic outcomes. The authors should give a more detailed discussion regarding this important method.

7, There are two Fig 1 and two Fig 4 in the current version, please check and revise. Also, there are errors in the reference section, please check the abbreviation of the journals, the authors should carefully revise them.

Round 2

Reviewer 2 Report

The authors have carefully revised the manuscript, I think the overall quality of the paper now meets the very high standard of this journal.